# Fairness in Aggregation: Optimal Top-$k$ and Improved Full Ranking

**Diptarka Chakraborty** [1]   **Arya Mazumdar** [2]   **Barna Saha** [2]   **Alvin Hong Yao Yan** [1]

## Abstract

Ensuring fairness in algorithmic ranking systems is a critical challenge with significant societal implications for hiring, recommendations, web search, and data management. Standard methods for aggregating multiple preference orders into a consensus ranking may perpetuate and even amplify the lack of representation of underrepresented groups. To address this, recent research has focused on incorporating fairness constraints to ensure the presence of different groups in the top-$k$ positions of the final aggregate ranking. We study two fairness-aware variants under the well-known Spearman footrule, which corresponds to the $L_1$ distance between rankings. First, we address the practically salient task of computing a fair aggregate top-$k$ ranking – crucial in settings like recommendations and hiring where selection is primarily based on the top-$k$ results – and present the first optimal algorithm for this problem. Second, we consider fair (full) rank aggregation over all candidates (not specifically on top-$k$). We already know of a 3-approximation for this fair rank aggregation variant (Wei et al., SIGMOD'22; Chakraborty et al., NeurIPS'22), whereas an exact algorithm exists for the corresponding unconstrained (unfair) version (Dwork et al., WWW'01). Closing the computational gap between fair and unconstrained rank aggregation has remained a tantalizing open problem. We make significant progress by giving a 2-approximation algorithm for fair (full) rank aggregation, improving substantially over the previous 3-approximation. Further, we complement our theoretical contributions with experiments on different real-world datasets, which corroborate our theoretical results and demonstrate strong empirical performance relative to state-of-the-art baselines.

The author order is alphabetical. [1]National University of Singapore [2]University of California San Diego. Correspondence to: Alvin Hong Yao Yan <alviny@u.nus.edu>.

*Proceedings of the 43$^{rd}$ International Conference on Machine Learning*, Seoul, South Korea. PMLR 306, 2026. Copyright 2026 by the author(s).

## 1. Introduction

The problem of aggregating multiple, often conflicting, preference orderings or rankings over a set of alternatives or candidates into a single consensus ordering, known as *rank aggregation*, is one of the most fundamental challenges, dating back to as early as the 18th century (Borda, 1781; Condorcet, 2014). This task is prevalent in numerous real-world scenarios, including hiring, recommendation systems, web searches, scholarship selections, and many more. The problem is often formulated as an optimization task over permutation/ranking metric space: given a set of rankings, find a ranking that minimizes the sum of distances to the input rankings. The ubiquity of this problem in scenarios requiring a selection or the generation of ranked suggestions has elicited considerable attention from the computational perspective (Dwork et al., 2001b; Fagin et al., 2003b; Gleich & Lim, 2011; Azari Soufiani et al., 2013; Li et al., 2017).

One of the classical distance metrics defined over a set of rankings is the *Spearman footrule* distance – it quantifies how far two rankings differ by summing absolute rank shifts across all candidates (Spearman, 1904; 1906), i.e., the rank-wise $L_1$ distance (see Section 2 for a formal definition). As an $L_1$-type measure, it emphasizes absolute positional disagreements and tends to be more responsive to large movements than counts of pairwise inversions. It is widely used in databases (Fagin et al., 2003b), clustering and analysis of ranked data (Marden, 1996), meta searches and learning to rank in information retrieval (Aslam & Montague, 2001; Liu, 2009), web searches (Dwork et al., 2001b), bioinformatics and genomics (Pihur et al., 2009) and social choice (Balinski & Laraki, 2011). Computationally, it is straightforward to evaluate. It also makes the rank aggregation problems tractable (Dwork et al., 2001a); in fact, aggregation under the footrule closely aligns with median-based aggregation, enabling simple heuristics that are effective in database settings (Fagin et al., 2003b). Computational tractability is a major advantage compared to some other relevant metrics on rankings, such as *Kendall–tau* – yet another classical distance metric. Despite its simplicity, the footrule is tightly related to *Kendall–tau* being equivalent up to a factor of two (Diaconis & Graham, 1977). Finally, in many practical contexts, such as hiring, the top-$k$ positions of a ranking are only (the most) relevant; unlike several other metrics, the footrule naturally extends to truncated lists and

accommodates missing ranks, further enhancing its utility in real-world scenarios (Fagin et al., 2003a).

This paper extends the classical rank aggregation task by incorporating additional constraints in the final output to ensure fair representations. The high-stakes nature of decisions made with ranking algorithms – such as offering employment, granting admissions to educational institutions, allocating loans, or prioritizing medical care during crises – underscores the need for proportional outcomes. Without explicit fairness considerations, these systems risk perpetuating systemic biases and harmful stereotypes against underrepresented groups, which are often defined by sensitive attributes like race, gender, or caste (Kay et al., 2015; Costello et al., 2016; Bolukbasi et al., 2016). In many countries, fairness constraints are enforced by legal norms to counteract historical discrimination, such as reservation systems for jobs and education in India (Borooah, 2010), affirmative actions in employment in South Africa (Burger et al., 2010; Thaver, 2017), and candidate selection by political parties in Spain (Verge, 2010).

To address the concerns on fairness and diversity in ranking algorithms, (Wei et al., 2022; Chakraborty et al., 2022) initiated the study of the rank aggregation problem considering the notion of proportional fairness (Baruah et al., 1996). This fairness criterion mandates that the representation of each protected group within the top-$k$ – often the most relevant – segment of the final ranking is appropriately balanced; more specifically, it satisfies (i) *minority protection*: each group has at least a certain (input specified) fraction of representative in the top-$k$ positions, and (ii) *restricted dominance*: none of the groups has more than a certain (input specified) fraction of representative in the top-$k$ positions (see Section 2 for a formal definition).

In the *fair rank aggregation* problem, given a set of input rankings over $d$ candidates, the objective is to find a fair ranking that minimizes the sum of distances to the input rankings. In this paper, we focus on the Spearman footrule distance. This problem was first explored by (Wei et al., 2022; Chakraborty et al., 2022), who proposed a simple baseline meta-algorithm. Their approach is to compute a closest fair ranking for each input permutation, then select the one with the minimum sum of distances to all inputs. Since a polynomial time algorithm for finding a closest fair ranking is already known (Celis et al., 2018), through a direct application of the triangle inequality, this meta-algorithm yields a 3-approximation. There is no better bound known for this fair variant until now, which is in stark contrast with the classical (unconstrained) rank aggregation problem that is solvable in polynomial time (Dwork et al., 2001a). Closing this complexity gap between the unconstrained and fair versions remains an important open problem, as also explicitly mentioned in (Wei et al., 2022).

For many real-world applications, such as hiring, where open positions are limited, the primary interest lies only in the top-$k$ portion of the final ranking. Furthermore, the requirements of many information retrieval applications necessitate a focus solely on the top-$k$ ranked items. Such applications motivate the introduction of *generalized Spearman footrule* to study top-$k$ lists (Fagin et al., 2003a). While the best current algorithm achieves a 3-approximation for the full ranking, it does not offer an explicit guarantee when attention is confined to the top-$k$ list/ranking. This naturally raises a key question: can we develop an algorithm with a similar or even better performance when only the top-$k$ positions are of interest?

**Our Contribution.** Our first main contribution is an algorithm to compute an optimal fair top-$k$ ranking, more specifically, an optimal $k$-fair top-$k$ ranking (see Definition 2.3).

**Theorem 1.1** (Informal). *There is a polynomial-time algorithm that, given a set of rankings over $d$ candidates and an integer $k$, computes an optimal aggregate $k$-fair top-$k$ ranking.*

Our primary contribution is a novel method for finding an optimal solution that perfectly adheres to the fairness constraints. To establish the above result, we first formulate the problem as an integer linear program (ILP) and consider its linear program (LP) relaxation. Typically, a natural next step would be to solve the LP and then round a fractional optimal solution. However, for problems with fairness constraints, such rounding often causes violations in fairness by up to a certain factor (e.g., (Bera et al., 2019; Chen et al., 2019; Ahmadian & Negahbani, 2023; Duppala et al., 2025)). To circumvent this issue of potential violation of the fairness constraint, we deviate from the usual rounding step. Instead, we study the constraint matrix and show that it is *totally unimodular* by leveraging its structural properties. This enables us to argue that the LP has an integral optimal solution, which we can obtain just by solving the LP (using the Ellipsoid method). We provide the details in Section 3. To the best of our knowledge, we are the first to use this novel technique for problems with fairness constraints.

Next, we focus on the fair (full) rank aggregation problem, where the task is to output a fair aggregate ranking over all the candidates (not just top-$k$). The unconstrained variant can be solved (exactly) in polynomial time by reducing it to a minimum cost perfect matching problem on a complete bipartite graph (Dwork et al., 2001a). By following the same reduction, it is not hard to observe that solving a fair variant of this minimum cost perfect matching problem suffices to get a solution to the fair rank aggregation problem. However, we show that such a fair matching problem, even on a complete bipartite graph, is NP-hard (see the Appendix D).

We then design an algorithm that attains a 2-approximation for the fair (full) rank aggregation problem, improving over the state-of-the-art 3-approximation (Wei et al., 2022; Chakraborty et al., 2022).

**Theorem 1.2** (Informal). *There is a polynomial-time algorithm that, given a set of rankings over $d$ candidates, computes a 2-approximate fair aggregate full ranking.*

To establish the above result, we start by solving the $k$-fair top-$k$ rank aggregation problem (with a slight modification in the objective as described in Section 4). Next, we extend the top-$k$ output ranking to a full ranking using a matching-based technique. The key novelty lies in our analysis, which shows that this extension yields a 2-approximation to the fair (full) rank aggregation problem.

To validate our approach, we conduct a comprehensive empirical study comparing our algorithm to baselines on several standard datasets. The experiments demonstrate that our method yields a substantially improved objective value (the sum of distances), outperforming state-of-the-art algorithms for fair rank aggregation. It is quite intriguing that, while we prove a 2-approximation guarantee via theoretical analysis, our algorithm consistently finds solutions that are nearly optimal in practice. Further, we design another 2-approximation algorithm (provided in the Appendix B), and when we return the best (in terms of objective value) of the outputs produced by these two 2-approximation algorithm, we attain further improvement in the objective in our empirical study. Another interesting finding is that our algorithm is *robust* to the choice of distance metric. Though as an immediate corollary of our theoretical result, we get a 4-approximation for the fair rank aggregation under the Kendall-tau metric (for the sake of completeness, we provide the proof in Appendix A), our empirical findings establish that our algorithm produces a fair ranking that is close (and in some cases even better) compared to the state-of-the-art fair rank aggregation algorithm with respect to the Kendall-tau metric due to (Chakraborty et al., 2025b).

**Related Works.** The rank aggregation problem, with and without fairness constraint, has been extensively studied (Kemeny, 1959; Young, 1988; Young & Levenglick, 1978; Dwork et al., 2001b; Ailon et al., 2008) with respect to different other distance metrics, such as Kendall-tau and Ulam. In the unconstrained setting, we know of a $(1 + \varepsilon)$-approximation (for any $\varepsilon > 0$) algorithm for the Kendall-tau (Kenyon-Mathieu & Schudy, 2007) and a 1.999-approximation for Ulam (Chakraborty et al., 2021; 2023). On the other hand, for the fairness-constrained variant, a recent result (Chakraborty et al., 2025b) shows a 2-approximation for Kendall-tau and slightly better than a 3-approximation for Ulam for a constant number of groups (Chakraborty et al., 2022). Apparently, there is no close relationship between the Spearman footrule distance and the Ulam distance. However, it is well-known (Diaconis & Graham, 1977) that the Spearman footrule distance between any two rankings is at least their Kendall-tau distance and at most twice their Kendall-tau distance. Consequently, the current best 2-approximation for fair rank aggregation under Kendall-tau (Chakraborty

et al., 2025b) implies a 4-approximation for the problem under Spearman footrule, a result that is weaker than the state-of-the-art 3-approximation (Chakraborty et al., 2022).

It is worth noting that apart from proportional fairness, other notions of fairness have also been explored in the literature. For instance, the notion of top-$k$ statistical parity and pairwise statistical parity has been considered in (Kuhlman & Rundensteiner, 2020). However, such a statistical notion of fairness can be quite restrictive and fails to satisfy the stronger guarantees of proportional fairness, as detailed in (Wei et al., 2022) with a concrete example.

In addition to rank aggregation, fairness has been incorporated into other ranking problems. (Celis et al., 2018) considered the problem of computing a nearest/closest proportional fair ranking to a given ranking across various metrics, including Spearman footrule, Bradley-Terry, and DCG. The concept of *robust fairness* has also been studied, where a close, fairer ranking is sought even without access to protected attributes (Kliachkin et al., 2024). It is crucial to note, however, that such algorithms are designed to adjust a single input and may not be effective for creating a good aggregate ranking – by first running any unconstrained rank aggregation algorithm and then making that output fair only leads to a 3-approximation (Chakraborty et al., 2022).

The rank aggregation problem can be interpreted as the median problem – namely, the 1-clustering problem – defined over a metric space of rankings. There is an expanding literature on clustering under proportional fairness, including work on $k$-clustering (Chierichetti et al., 2017; Huang et al., 2019), proportional clustering (Chen et al., 2019), fair representational clustering (Bera et al., 2019; Bercea et al., 2019), pairwise fair clustering (Bandyapadhyay et al., 2024; 2025), correlation clustering (Ahmadian et al., 2020; Ahmadi et al., 2020; Ahmadian & Negahbani, 2023; Chakraborty et al., 2026a), and consensus clustering (Chakraborty et al., 2025a; 2026a;b), among others. However, it should be noted that the notion of fairness in general clustering differs from that in rank aggregation.

## 2. Preliminaries

**Notations.** For any $n \in \mathbb{N}$, let $[n]$ denote the set $\{1, 2, \cdots, n\}$. We refer to the set of all rankings (or permutations) over $[d]$ by $\mathcal{S}_d$. For an element $a \in [d]$, we denote its rank in $\pi$ by $\pi(a)$. For a ranking $\pi$, we say that it ranks $a$ ahead of $b$ if $\pi(a) < \pi(b)$. For a two-dimensional matrix $A$ and integers $i, j$, we use $A_{ij}$ to denote the $(i, j)$-th entry of $A$. We use $\mathbb{1}_E$ to denote an indicator function for an event $E$.

In this paper, we use the following definition of *fair ranking* from (Chakraborty et al., 2022).[1]

---

[1]Note, a similar definition was considered in (Wei et al., 2022),

**Definition 2.1** (Fair Ranking). Consider a partitioning of $d$ candidates into $g$ groups $G_1, \cdots, G_g$, and for each group $G_i$, $i \in [g]$, parameters $\alpha_i, \beta_i \in [0,1]$. For $\bar{\alpha} = (\alpha_1, \cdots, \alpha_g)$, $\bar{\beta} = (\beta_1, \cdots, \beta_g)$, and $k \in [d]$, a ranking $\pi$ on (a subset of) $d$ candidates is called $(\bar{\alpha}, \bar{\beta})$-*k-fair* if for each group $G_i$:

- *Minority Protection*: The top-$k$ positions of $\pi$ contain at least $\lfloor \alpha_i \cdot k \rfloor$ candidates from $G_i$, and

- *Restricted Dominance*: The top-$k$ positions of $\pi$ contain at most $\lceil \beta_i \cdot k \rceil$ candidates from $G_i$.

When clear from the context, we omit $\bar{\alpha}, \bar{\beta}$, and $k$ from the notation, and simply refer to it as *fair ranking*.

**Distance metric.** For any two rankings $\pi_1, \pi_2 \in \mathcal{S}_d$, the *Spearman footrule* distance between them, denoted by $F(\pi_1, \pi_2)$, is the sum of the displacement of elements between $\pi_1$ and $\pi_2$; formally $F(\pi_1, \pi_2) := \sum_{i \in [d]} |\pi_1(i) - \pi_2(i)|$.

Instead of a full ranking, we are often interested in a *top-k list* or *top-k ranking*, which ranks only a subset of $k$ (top-most) candidates, and ignores the rest. Formally, a *top-k ranking* $\tau$ is a bijection from a domain $D_\tau \subseteq [d]$ to $[k]$, where $D_\tau$ denotes the set of candidates ranked by $\tau$. We use the following generalization of Spearman footrule distance from (Fagin et al., 2003a) to handle top-$k$ rankings.

For a (full) ranking $\sigma \in \mathcal{S}_d$ and a top-$k$ ranking $\tau$, their Spearman footrule distance is defined as

$$F(\sigma, \tau) := \sum_{i \in D_\tau} |\sigma(i) - \tau(i)| + \sum_{i \in [d] \setminus D_\tau} |\sigma(i) - (k+1)|.$$

We abuse the notation to use $F(\cdot, \cdot)$ to denote Spearman footrule distance between two full rankings as well as between one full ranking and another top-$k$ ranking.

**Fair Rank Aggregation.**

**Definition 2.2** (Fair Rank Aggregation). Given a set $S \subseteq S_d$ of rankings over $d$ candidates that are partitioned into $g$ groups $G_1, \cdots, G_g$, $\bar{\alpha} = (\alpha_1, \cdots, \alpha_g) \in [0,1]^g$, $\bar{\beta} = (\beta_1, \cdots, \beta_g) \in [0,1]^g$, and $k \in [d]$, the *fair rank aggregation* problem seeks to find a $(\bar{\alpha}, \bar{\beta})$-$k$-fair ranking $\sigma \in S_d$ that minimizes the objective function $\mathrm{Obj}(S, \sigma) := \sum_{\pi \in S} F(\pi, \sigma)$.

We also consider a variant where, instead of requiring a full ranking over all $d$ candidates, only a top-$k'$ ranking/list is desired. Our definition of fair ranking (Definition 2.1) naturally extends to a top-$k'$ ranking (for $k' \geq k$) – a $(\bar{\alpha}, \bar{\beta})$-$k$-fair top-$k'$ ranking $\sigma$ satisfies the fairness constraints (minority protection and restricted dominance) in its top-$k$ ranked candidates.

**Definition 2.3** ($k$-Fair Top-$k'$ Rank Aggregation). Given a set $S \subseteq S_d$ of rankings over $d$ candidates that are

however it was slightly restrictive.

partitioned into $g$ groups $G_1, \cdots, G_g$, $\bar{\alpha} = (\alpha_1, \cdots, \alpha_g) \in [0,1]^g$, $\bar{\beta} = (\beta_1, \cdots, \beta_g) \in [0,1]^g$, and $k \in [d]$, the *k-fair top-k' rank aggregation* problem asks to find a $(\bar{\alpha}, \bar{\beta})$-$k$-fair top-$k'$ ranking $\tau$ that minimizes $\mathrm{Obj}(S, \tau) := \sum_{\pi \in S} F(\pi, \tau)$.

In this paper, we provide an exact algorithm for the special case of $k$-fair top-$k$ rank aggregation, i.e., when the value of $k$ is identical for the fairness constraint and ranking output (more specifically, $k' = k$). For brevity, we refer to this as *fair top-k rank aggregation*. We also study the special case of $k' = d$ that corresponds to requiring a full ranking as output, and is the fair rank aggregation problem as defined in Definition 2.2.

For brevity, when $S$ is clear from the context, we omit it from the notation of the objective function and simply use $\mathrm{Obj}(\tau)$ to denote $\mathrm{Obj}(S, \tau)$.

## 3. Fair Top-$k$ Rank Aggregation

In this section, we show one of our main results: the problem of $k$-fair top-$k$ rank aggregation can be solved exactly in polynomial time.

**Theorem 3.1.** *There exists a polynomial-time algorithm that, given a set $S \subseteq S_d$ of rankings over $d$ candidates that are partitioned into $g$ groups $G_1, \cdots, G_g$, $\bar{\alpha}, \bar{\beta} \in [0,1]^g$, and $k \in [d]$, finds an optimal $(\bar{\alpha}, \bar{\beta})$-k-fair top-k aggregate ranking.*

**ILP formulation.** To show the above result, we start by formulating the $k$-fair top-$k$ rank aggregation problem using an integer linear program (ILP). For that purpose, we first define the following weight function: For each $i \in [d]$ and positive integer $j \leq (k+1)$, the weight $w_{ij}$ is equal to the distance incurred by placing element $i$ at position $j$ in the output ranking, more specifically, $w_{ij} = \sum_{\pi \in S} |\pi(i) - j|$. Next, we consider the variable $x_{ij}$ for each $i \in [d]$ and positive integer $j \leq k$, where $x_{ij}$ takes on 1 if element $i$ is placed at position $j$; and 0 otherwise.

It is now easy to observe that the following ILP formulates the top-$k$ fair rank aggregation problem.

$$
\begin{aligned}
&\text{minimize } \sum_{i \in [d]} \sum_{j \leq k} w_{ij} x_{ij} \\
&+ \sum_{i \in [d]} (1 - \sum_{j \leq k} x_{ij}) w_{i(k+1)} \\
&\text{subject to} \\
&\sum_{j \leq k} x_{ij} \leq 1 && \forall i \in [d] && (1) \\
&\sum_{i \in [d]} x_{ij} = 1 && \forall j \leq k && (2) \\
&\sum_{i \in G_a} \sum_{j \leq k} x_{ij} \geq \lfloor \alpha_a \cdot k \rfloor && \forall G_a \in \mathcal{G} && (3) \\
&\sum_{i \in G_a} \sum_{j \leq k} x_{ij} \leq \lceil \beta_a \cdot k \rceil && \forall G_a \in \mathcal{G} && (4) \\
&x_{ij} \in \{0,1\} && \forall i \in [d], j \leq k && (5)
\end{aligned}
$$
$$\text{(Fair-K-LP)}$$

Next, we relax the integrality constraint on $x_{ij}$ to replace the $0 - 1$ constraint with $0 \leq x_{ij} \leq 1$ to get the corresponding linear program (LP).

We know that if an LP satisfies certain properties, the polyhedron defined by the constraints is *integral* – the LP

will be integral, i.e., it will have at least an integral optimal solution (if an optimum exists). In that case, it suffices to solve the LP to derive an integral optimal solution. One such condition is that when expressed in *standard form*, $A\bar{x} = \bar{b}, \bar{x} \geq 0$ for a matrix $A$ and vectors $\bar{x}, \bar{b}$, the matrix $A$ is *totally unimodular* (see Theorem 3.2). We also refer to any standard textbook on LP (e.g., (Schrijver, 1998; Wolsey & Nemhauser, 1999)) for well-known notions such as standard form, total unimodularity, and integrality of an LP.

**Theorem 3.2** ((Wolsey & Nemhauser, 1999), Part III.I Proposition 1.3, 2.3)**.** *Let $A$ be a totally unimodular matrix, and $\bar{b}$ be an integral vector. Then the polyhedron $P(\bar{b}) = \{\bar{x} \in \mathbb{R}^n_+ : A\bar{x} = \bar{b}\}$ is integral. Furthermore, the linear program $\{\min \bar{c}^T \bar{x} : \bar{x} \in P(\bar{b})\}$ has an integral optimal solution for all $\bar{c} \in \mathbb{Z}^n$ for which an optimal solution exists.*

Now, we prove that when expressed in linear form, the constraints of Fair-K-LP form a totally unimodular matrix. The following theorem is a known characterization of totally unimodular matrices, which we use.

**Theorem 3.3** ((Wolsey & Nemhauser, 1999), Part III.I Theorem 2.7)**.** *A matrix $A$ is totally unimodular if and only if the following holds. For every subset $R$ of the rows of $A$, there is a partitioning of $R$ into $R_1$ and $R_2$ such that for every column $c$ of the matrix $A$, $\left|\sum_{r \in R_1} A_{rc} - \sum_{s \in R_2} A_{sc}\right| \leq 1$.*

Before we prove that the matrix for Fair-K-LP is totally unimodular, we can make some observations about its standard form. Each variable $x_{ij}$ only appears in exactly one constraint of (1), (2), (3), (4), and (5) each. Therefore, ignoring slack variables, any column of $A$ has exactly five non-zero entries.

**Lemma 3.4.** *Let the constraints of the linear program Fair-K-LP be expressed in standard form as $A\bar{x} = \bar{b}$. Then the matrix $A$ is totally unimodular.*

*Proof.* Let $R$ be any subset of the rows of $A$. Now, our goal will be to form a partitioning of $R$ into two subsets, $R_1$ and $R_2$ such that for every column $c$, $\left|\sum_{i \in R_1} A_{ic} - \sum_{i \in R_2} A_{ic}\right| \leq 1$.

Let us first make some observations to simplify the proof. Suppose both rows corresponding to constraints of (3) and (4) for the same group are present in $R$. Then, for any column, either of the two cases holds for these two rows: (i) both entries are zero, (ii) one entry is $-1$ and one entry is 1. Therefore, we can easily place both rows into the same partition, and the sum over these two rows for all columns is 0. Therefore, without loss of generality, we will assume that at most one of the two fairness constraints, i.e., (3) and (4), for each group is present in $R$.

Suppose only the row corresponding to the constraint of (3) on a group is present in $R$. Observe that it has an entry of $-1$ in every column where the constraint of (4) has a 1 in. If we place (3) into a partition, say $R_1$, then the impact on $\left|\sum_{i \in R_1} A_{ic} - \sum_{i \in R_2} A_{ic}\right|$ is equivalent for all columns $c$ as

placing the row (4) for this group in $R_2$. So without loss of generality, we will assume that no rows corresponding to (3) are present in $R$. If any such row is present, then it is equivalent to replacing it by the row corresponding to (4).

Further, any row corresponding to a constraint of (5) has an entry of 1 only in a single column, say $c$. For all such rows in $R$, we can 'defer' their placement into the partitioning until last, where they can easily be placed in the partition for which $c$ has a smaller sum. Therefore, we will also assume any rows corresponding to a constraint of (5) do not appear in $R$.

Now we describe the partitioning $R_1$ and $R_2$. For all rows in $R$ corresponding to a constraint of (4), place them into $R_1$. For all rows in $R$ corresponding to a constraint of (2), place them into $R_2$. Now, consider rows in $R$ corresponding to a constraint of (1). For some such row $r$, the constraint corresponding to this row must be for some $i \in [d]$. Let $G_a$ be the group that $i$ belongs to. If the row corresponding to the fairness constraint for $G_a$ is in $R$, then place $r$ into $R_2$. Otherwise, place it into $R_1$.

Recall that our goal is for any column $c$, $\left|\sum_{i \in R_1} A_{ic} - \sum_{i \in R_2} A_{ic}\right| \leq 1$. We now prove this holds for the given partitioning. This column $c$ contains the coefficients of some variable, say $x_{ij}$. Therefore, we know that there must exist three rows of $A$ which contain a 1 in this column, corresponding to a constraint, each of type (1), (2), and (4). Now, let us consider the partitioning scheme under all possible subsets of these four rows being in $R$.

First, if at most one of the rows is present in $R$, it is obvious that $\left|\sum_{i \in R_1} A_{ic} - \sum_{i \in R_2} A_{ic}\right| = 1$.

Now consider the case that exactly two of the rows corresponding to constraints of (1), (2), or (4) are present in $R$. There are three such possibilities. Under all three such possibilities, it is easy to verify that our described partitioning indeed guarantees that one such row is placed in $R_1$, and another row is placed in $R_2$.

Finally, consider the case that all three of the rows corresponding to constraints of (1), (2), or (4) are present in $R$. Then by our partitioning scheme, we have that $\sum_{i \in R_1} A_{ic} = 1$ as the row corresponding to (4) is placed in $R_1$, and $\sum_{i \in R_2} A_{ic} = 2$ as the row corresponding to (1) and (2) are placed in $R_2$. Therefore, the difference in the sums is 1.

Now, we have shown that for every subset $R$ of the rows of $A$, there exists a partitioning of $R$ into $R_1$ and $R_2$ such that $\left|\sum_{i \in R_1} A_{ij} - \sum_{i \in R_2} A_{ij}\right| \leq 1$ holds. By Theorem 3.3, the matrix $A$ is totally unimodular. $\square$

Finally, we find an optimal basic feasible solution to the linear program in polynomial time using the ellipsoid method.

**Theorem 3.5** (Theorem 3.8 of (Grötschel et al., 1981))**.** *There exists an algorithm that, given a matrix $A \in \mathbb{Q}^{m \times n}$ and*

vectors $\bar{b} \in \mathbb{Q}^n, \bar{c} \in \mathbb{Q}^m$, finds $\bar{x}$ that maximizes $\bar{c}^T \bar{x} : A\bar{x} \leq \bar{b}$. Further, $\bar{x}$ is a vertex of the polyhedron $P(\bar{b}) = \{\bar{x} \in \mathbb{R}^n_+ : A\bar{x} \leq \bar{b}\}$. Moreover, the algorithm runs in time polynomial in $n$ and $\log L$, where $L$ is an upper bound on the absolute values of the numerators and denominators of all entries of $A, \bar{b}, \bar{c}$.

*Proof of Theorem 3.1.* We have from Lemma 3.4 that the matrix $A$ corresponding to the linear program is totally unimodular. Further, it is obvious that the vector $\bar{b}$ of the linear program is integral. Then by the properties of totally unimodularity (Theorem 3.2), and by the fact that a nonempty polyhedron is integral if and only if all of its extreme points are integral (Wolsey & Nemhauser, 1999), all basic feasible solutions of Fair-K-LP are integral. By using an ellipsoid method and using a perturbation on the objective function (as necessary by Theorem 3.5), such an optimal basic feasible solution can be found in polynomial time. □

# 4. Approximate Fair Full Rank Aggregation

In this section, we design an approximation algorithm for the fair (full) rank aggregation problem. Namely, we prove the following theorem.

**Theorem 4.1.** *There exists a polynomial-time algorithm that, given a set $S \subseteq S_d$ of rankings over $d$ candidates that are partitioned into $g$ groups $G_1, \cdots, G_g$, $\bar{\alpha}, \bar{\beta} \in [0,1]^g$, and $k \in [d]$, finds a 2-approximate $(\bar{\alpha}, \bar{\beta})$-k-fair aggregate ranking on $d$ candidates.*

**Extending a Top-$k$ list to a full ranking.** To show this result, we first define a subproblem of extending a top-$k$ ranking to a full ranking.

**Definition 4.2** (Minimum Cost Top-$k$ List Completion Problem). We are given $S \subseteq S_d$ and a top-$k$ list $\tau$. Define $\tilde{S}_d := \{\sigma \in S_d \mid \forall a \in D_\tau, \sigma(a) = \tau(a)\}$. Then the *minimum cost top-$k$ list completion* problem is to find a $\sigma \in \tilde{S}_d$ that minimizes $2\sum_{\pi \in S}\sum_{i \in [d]}(\pi(i) - \sigma(i)) \cdot \mathbb{1}_{\sigma(i) < \pi(i)}$.

**Lemma 4.3.** *There is an algorithm that, given an instance ($S \subseteq S_d$ and a top-k list $\tau$) of the minimum cost top-k list completion problem, finds an optimal solution in time $O(nd^2 + d^3)$, where $n = |S|$.*

*Proof.* We reduce this to an instance of minimum cost perfect matching as follows. Form a set $V$ by creating a vertex for each $a \in [d] \setminus D_\tau$, and a set $W$ by creating a vertex for each $b \in \{k+1, \cdots, d\}$. For each $a \in V, b \in W$, add an edge with weight equal to $2\sum_{\pi \in S}(\pi(a) - b) \cdot \mathbb{1}_{b < \pi(a)}$, and let the set of edges be $E$. It is straightforward to see that $G = (V \cup W, E)$ forms a weighted complete bipartite graph. See Figure 1 for an illustration given $S$ consists of $\pi_1 = \{1,2,3,4,5,6\}, \pi_2 = \{3,6,5,1,2,4\}, \pi_3 = \{6,1,2,5,4,3\}$

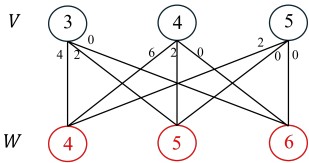

*Figure 1.* Example of the graph generated by the reduction for the example in the proof of Lemma 4.3. For instance, the cost of placing 3 at rank 4 is equal to 4, as $\pi_1(3) < 4, \pi_2(3) < 4$ and $\pi_3(3) - 4 = 2$.

and $\tau = \{1,2,6\}$. It is well-known that the problem of finding a minimum cost perfect matching in a complete bipartite graph can be solved in time $O(v^3)$ (Edmonds & Karp, 1972) where $v$ is the number of vertices in the graph. Given a minimum cost perfect matching, $\sigma$ can easily be constructed by iterating over the matching, and for each matched edge $(a,b)$, placing the element $a$ at rank $b$ in $\sigma$.

As the number of vertices in the constructed graph $G$ is at most $2d$, and the reduction can be done in time $O(nd^2)$, the running time is bounded by $O(nd^2 + d^3)$. □

**Fair Rank Aggregation.** Our algorithm (to prove Theorem 4.1) proceeds in two steps. It first constructs a top-$k$ list, and then extends it to a full ranking. We first describe a polynomial-time algorithm $\mathcal{A}$ to obtain this top-$k$ list. Consider the problem of finding a fair top-$k$ list $\tau$ which minimizes the following objective: $2\sum_{\pi \in S}\sum_{i \in D_\tau}(\pi(i) - \tau(i)) \cdot \mathbb{1}_{\tau(i) < \pi(i)}$.

Observe that this can be formulated as a linear program where all constraints are identical to Fair-K-LP (Section 3), except that the objective is to minimize $\sum_{i \in [d]}\sum_{j \leq k} w_{ij} x_{ij}$, where $w_{ij} = 2\sum_{\pi \in S}(\pi(i) - j) \cdot \mathbb{1}_{j < \pi(i)}$. As the constraints are identical, the proof of Lemma 3.4 holds identically for this linear program. By using a proof identical to that of Theorem 3.1, there exists an algorithm that outputs a fair top-$k$ list $\tau$ that minimizes the above objective in polynomial time. Let $\mathcal{A}$ be this algorithm.

**Description of the algorithm.** Suppose we are given a set $S$ of rankings over $[d]$, of size $n$. Then we construct a ranking using the following two-step procedure.

First, apply algorithm $\mathcal{A}$ and obtain a fair top-$k$ ranking $\tau$. We then apply the algorithm from Lemma 4.3 to extend $\tau$ to obtain a full ranking $\sigma$. A formal description of the algorithm is given in Algorithm 1.

---

**Algorithm 1** FAIR RANK AGGREGATION

1: **Input:** Set of rankings $S \subseteq \mathcal{S}_d$
2: $\tau \leftarrow$ top-$k$ fair list by applying algorithm $\mathcal{A}$ on $S$.
3: $\sigma \leftarrow$ full ranking from using Lemma 4.3 with $S$ and $\tau$ as input.
4: Return $\sigma$

---

**Analysis of the algorithm.** Let us start by making some useful observations that we will use in the analysis. Recall that the Spearman footrule distance between two rankings is equal to the sum of displacements between the elements. It must be that the sum of rightward displacements must equal the sum of leftward displacements (Mathieu & Mauras, 2020). Formally,

**Proposition 4.4.** *Let* $\pi, \sigma \in \mathcal{S}_d$. *Then it holds that* $\sum_{i \in [d]} (\sigma(i) - \pi(i)) \cdot \mathbb{1}_{\pi(i) < \sigma(i)} = \sum_{i \in [d]} (\pi(i) - \sigma(i)) \cdot \mathbb{1}_{\pi(i) > \sigma(i)}.$

Let $L \subseteq [d]$ be the set of elements placed in the top-$k$ positions in $\sigma$ and $R = [d] \setminus L$. Let $\sigma^*$ denote an (arbitrary) optimal fair aggregate ranking, $L^* \subseteq [d]$ be the set of elements placed in its top-$k$ positions, and $R^* = [d] \setminus L^*$.

From Proposition 4.4 we have that $F(\pi, \sigma) = 2 \sum_{i \in [d]} (\pi(i) - \sigma(i)) \cdot \mathbb{1}_{\sigma(i) < \pi(i)}$. Therefore, it holds that $\mathtt{Obj}(\sigma) = 2 \sum_{\pi \in S} \sum_{i \in [d]} (\pi(i) - \sigma(i)) \cdot \mathbb{1}_{\sigma(i) < \pi(i)}$. Let us now define for any ranking $\sigma$ and $D \subseteq [d]$,

$$\overleftarrow{\mathtt{Obj}}(\sigma_D) := 2 \sum_{\pi \in S} \sum_{i \in D} (\pi(i) - \sigma(i)) \cdot \mathbb{1}_{\sigma(i) < \pi(i)}. \quad (6)$$

**Lemma 4.5.** *Consider the output ranking $\sigma$ in Algorithm 1. It holds that* $\mathtt{Obj}(\sigma) \leq \overleftarrow{\mathtt{Obj}}(\sigma^*_{L^*}) + OPT.$

*Proof.* The top-$k$ ranks of $\sigma$ is equivalent to $\tau$ that is the output of algorithm $\mathcal{A}$, which finds a top-$k$ list that minimizes $2 \sum_{\pi \in S} \sum_{i \in [d]} (\pi(i) - \tau(i)) \cdot \mathbb{1}_{\tau(i) < \pi(i)}$. Therefore,

$$\overleftarrow{\mathtt{Obj}}(\sigma_L) \leq \overleftarrow{\mathtt{Obj}}(\sigma^*_{L^*}). \quad (7)$$

Let us next analyze $\overleftarrow{\mathtt{Obj}}(\sigma_R)$. For the sake of analysis, consider a ranking $\tilde{\sigma}$ that places elements of $R$ in the ranks $k+1, \cdots, d$ in the following way. For any $a \in R \cap R^*$, set $\tilde{\sigma}(a) = \sigma^*(a)$, that is, place $a$ in the same rank as $\sigma^*$ does. Then for any $a \in R \setminus R^*$, place $a$ arbitrarily into any of the remaining available ranks.

By definition, $\overleftarrow{\mathtt{Obj}}(\tilde{\sigma}_R) = \overleftarrow{\mathtt{Obj}}(\tilde{\sigma}_{R \setminus R^*}) + \overleftarrow{\mathtt{Obj}}(\tilde{\sigma}_{R \cap R^*})$. As all the elements in $R \cap R^*$ are placed in the same ranks as they are in $\sigma^*$, clearly

$$\overleftarrow{\mathtt{Obj}}(\tilde{\sigma}_{R \cap R^*}) = \overleftarrow{\mathtt{Obj}}(\sigma^*_{R \cap R^*}) \leq \overleftarrow{\mathtt{Obj}}(\sigma^*_{R^*}). \quad (8)$$

Next, consider any element $a \in R \setminus R^*$. Observe that any such element must also be in $L^*$, and therefore be placed in the top-$k$ ranks of $\sigma^*$. This implies that $\sigma^*(a) < \tilde{\sigma}(a)$. Therefore for any $\pi \in S$,

$$(\pi(a) - \tilde{\sigma}(a)) \cdot \mathbb{1}_{\tilde{\sigma}(a) < \pi(a)} \leq (\pi(a) - \sigma^*(a)) \cdot \mathbb{1}_{\sigma^*(a) < \pi(a)}.$$

That is, the leftward displacement of $a$ in $\tilde{\sigma}$ is no more than the leftward displacement of $a$ in $\sigma^*$. By summing over all

rankings in $S$, all $a \in R \setminus R^*$ and multiplying both sides by a factor of 2,

$$2 \sum_{\pi \in S} \sum_{a \in R \setminus R^*} (\pi(a) - \tilde{\sigma}(a)) \cdot \mathbb{1}_{\tilde{\sigma}(a) < \pi(a)}$$
$$\leq 2 \sum_{\pi \in S} \sum_{a \in R \setminus R^*} (\pi(a) - \sigma^*(a)) \cdot \mathbb{1}_{\sigma^*(a) < \pi(a)}.$$

Thus, by Equation 6,

$$\overleftarrow{\mathtt{Obj}}(\tilde{\sigma}_{R \setminus R^*}) \leq \overleftarrow{\mathtt{Obj}}(\sigma^*_{R \setminus R^*})$$
$$\leq \overleftarrow{\mathtt{Obj}}(\sigma^*_{L^*}) \, (\text{As } R \setminus R^* \subseteq L^*). \quad (9)$$

Therefore, we get that

$$\overleftarrow{\mathtt{Obj}}(\tilde{\sigma}_R) = \overleftarrow{\mathtt{Obj}}(\tilde{\sigma}_{R \setminus R^*}) + \overleftarrow{\mathtt{Obj}}(\tilde{\sigma}_{R \cap R^*})$$
$$\leq \overleftarrow{\mathtt{Obj}}(\sigma^*_{L^*}) + \overleftarrow{\mathtt{Obj}}(\tilde{\sigma}_{R \cap R^*}) \, (\text{By Equation 9})$$
$$\leq \overleftarrow{\mathtt{Obj}}(\sigma^*_{L^*}) + \overleftarrow{\mathtt{Obj}}(\sigma^*_{R^*}) \leq OPT. \quad (10)$$

Since the ranking $\sigma$ is obtained by optimally solving the minimum cost top-$k$ ranking completion problem from $\tau$,

$$\overleftarrow{\mathtt{Obj}}(\sigma_R) \leq \overleftarrow{\mathtt{Obj}}(\tilde{\sigma}_R). \quad (11)$$

By combining the above inequalities, we obtain

$$\mathtt{Obj}(\sigma) = \overleftarrow{\mathtt{Obj}}(\sigma_{[d]})$$
$$= \overleftarrow{\mathtt{Obj}}(\sigma_L) + \overleftarrow{\mathtt{Obj}}(\sigma_R)$$
$$\leq \overleftarrow{\mathtt{Obj}}(\sigma_L) + \overleftarrow{\mathtt{Obj}}(\tilde{\sigma}_R) \quad (\text{By Equation 11})$$
$$\leq \overleftarrow{\mathtt{Obj}}(\sigma_L) + OPT \quad (\text{By Equation 10})$$
$$\leq \overleftarrow{\mathtt{Obj}}(\sigma^*_{L^*}) + OPT \quad (\text{By Equation 7}).$$

$\square$

*Proof of Theorem 4.1.* Consider the ranking $\sigma$ obtained in Algorithm 1. Next, from Lemma 4.5,

$$\mathtt{Obj}(\sigma) \leq \overleftarrow{\mathtt{Obj}}(\sigma^*_{L^*}) + OPT \leq \overleftarrow{\mathtt{Obj}}(\sigma^*_{[d]}) + OPT$$
$$\leq OPT + OPT = 2OPT.$$

Further, it is immediate from Theorem 3.1 and Lemma 4.3 that Algorithm 1 runs in polynomial time. $\square$

## 5. Experiments

In this section, we provide an empirical evaluation of our algorithm on real-world datasets. The algorithms are implemented using Python 3.12. Experiments are run on a laptop running Windows 11, using an Intel 275HX processor and 32GB of RAM. To find the optimal solution for comparison, we use Integer Linear Programming (ILP),

implemented using Gurobi 12.0.3 as the solver. The code and dataset can be found on Github[2].

**Baselines.** We use two baseline algorithms for comparison. First, the meta-algorithm of (Chakraborty et al., 2022), which finds a closest fair ranking for each input ranking, then returns the ranking among these that minimizes the objective cost. We label this algorithm "BFI" and it is known to be a 3-approximation. Second, we use the state-of-the-art algorithm for fair rank aggregation under the Kendall-tau distance by (Chakraborty et al., 2025b). Their efficiently implementable algorithm gives an $18/7$-approximation for fair rank aggregation under the Kendall-tau distance, and thus implying $36/7$-approximation under the Spearman footrule. We label this algorithm as "KT".

**Datasets.** We use two datasets previously introduced for fair rank aggregation. The first dataset was introduced by (Kuhlman & Rundensteiner, 2020) and consists of performance rankings provided by (real) experts over football players, taken from a fantasy sports website for American football. The dataset consists of 25 experts providing rankings over a set of players for 16 weeks of the 2019 season. We follow their work and divide players into two groups based on the conference the player's team is in. The second was introduced by (Wei et al., 2022) and consists of rankings by 7 users over 268 movies taken from the Movielens dataset. The movies belong to 8 genres, and we follow their work to place movies into groups based on genre. In all instances, we set the $\bar{\alpha},\bar{\beta}$ values to be equal to the proportion of each group in the input instance as an intuitive proportional fair constraint, same as in (Chakraborty et al., 2025b).

**Research Questions.** We also experimentally evaluate the solution quality of the proposed algorithm against the baselines.

RQ1: Is our algorithm able to find a fair aggregate ranking with a smaller objective cost than the current best algorithms for fair rank aggregation? How does it perform with respect to the theoretical guarantees of Theorem 4.1?

RQ2: How robust is our algorithm with respect to the distance metric? Is our algorithm able to find good approximate fair rankings under the Kendall-tau distance in practice?

**Results: RQ1.** We evaluate our algorithms against the baselines for the Movielens dataset with results in Figure 2 and for the football dataset with results in Figure 3. We implement Algorithm 1 labeled as such, and we further implement a slightly more intricate Algorithm 3, which is guaranteed to find a ranking with cost no more than Algorithm 1 (see Appendix C for details), albeit with the same 2-approximation in the theoretical analysis. As both algorithms find rankings with the same objective cost for the football dataset, we keep

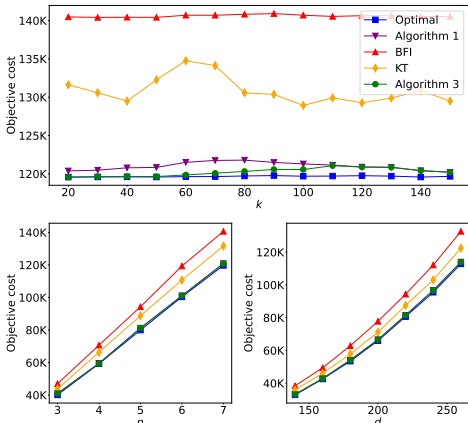

*Figure 2.* Results on Movielens dataset. Top figure shows varying $k$, bottom left shows varying $n$ and bottom right shows varying $d$.

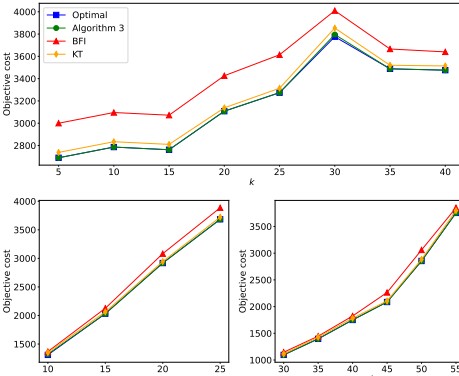

*Figure 3.* Results on football dataset (week 4). Top figure shows varying $k$, bottom left shows varying $n$ and bottom right shows varying $d$.

only Algorithm 3 in the figure for readability. As the results across all 16 instances of the football dataset are similar, we only plot results for one instance (week 4) of the dataset. We vary the parameters $k$, $n$, and $d$ in the input instances to study the impact on algorithm performance, varying $n$ and $d$ by deterministically keeping an (arbitrary) subset of the input instance (we observe the choice of subset does not affect the performance, and thus we plot for one such deterministically chosen arbitrary choice). For the Movielens dataset in varying $k$, we use the full instance ($n = 7, d = 268$), and for varying $n$ and $d$ we fix $k = d/2$. We do the same for the football dataset, using the full instance ($n = 25, d = 57$) for varying $k$, and fixing $k = d/2$ when varying $n$ and $d$.

We observe that our algorithm (Algorithm 3) performs much better than the theoretical guarantees. It finds a fair aggregate ranking with an objective cost within 2% - 3% from optimal in the Movielens dataset, and at most 1% from optimal in the football dataset (and often finds an optimal fair aggregate ranking). It also performs better than both baselines in both datasets. In the Movielens dataset, our algorithm finds a

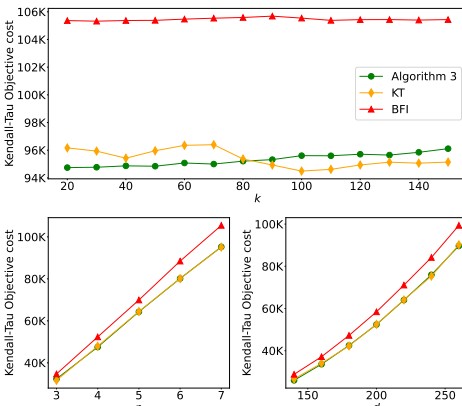

**Figure 4.** Results on the Movielens dataset for fair rank aggregation under the Kendall-tau distance.

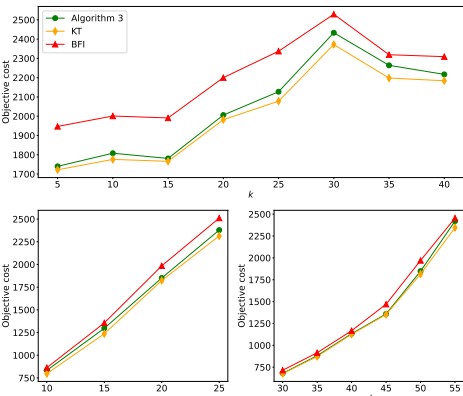

**Figure 5.** Results on football dataset for fair rank aggregation under Kendall-tau distance.

ranking with an objective cost 5% - 11% less than that found by KT, and 13% - 15% less than BFI. In the football dataset, our algorithm finds a ranking with an objective cost between 1% - 2% less than KT, and 5% - 10% less than BFI. Our algorithm is robust and performs well in all cases as the various parameters are varied.

**Results: RQ2.** We also test the efficiency of our algorithm on fair rank aggregation under the Kendall-tau metric. Note, as we have observed, Algorithm 3 provides slightly better empirical performance, so we implement and plot only this algorithm for all experiments for fair rank aggregation under the Kendall-tau distance. Recall that the Spearman footrule and Kendall-tau distance between two rankings are always within a factor of two (Diaconis & Graham, 1977). Therefore, Theorem 4.1 immediately implies as a corollary that our algorithm is a 4-approximation for fair rank aggregation under the Kendall-tau distance. We compare this algorithm to KT and BFI in the Movielens dataset, and plot the results in Figure 4. Our algorithm performs much better than the theoretical 4-approximation; in fact, it always outperforms the 3-approximation algorithm BFI. In particular, we observe

that our algorithm finds a ranking with objective cost within 2% of that found by KT. In fact, it occasionally finds a better fair aggregate ranking than KT. Despite BFI also being a 3-approximation algorithm in theory, our algorithm performs much better than BFI, finding a fair aggregate ranking with around 8% - 10% less objective cost than BFI.

We bolster our empirical findings on the performance of our algorithm for fair rank aggregation under the Kendall-tau metric by studying the results on the football dataset as well, and plot the results in figure 5. We observe that our algorithm performs consistently slightly worse than KT (which is also expected) in all instances on this dataset. However, it still finds a fair aggregate ranking with cost at most 2% worse than that found by KT. Our algorithm continues to perform significantly better than BFI in practice, finding a fair aggregate ranking between 4% to 10% better than that of BFI across all instances.

## 6. Conclusion and Future Work

This paper investigates the problem of fair rank aggregation, specifically for the top-$k$ and full rank variants, under the Spearman footrule metric. Incorporating the proportional fairness constraint, we propose an exact algorithm for the top-$k$ problem and a 2-approximation algorithm for the full rank version, which improves upon the state-of-the-art 3-approximation (Wei et al., 2022; Chakraborty et al., 2022).

Empirical evaluation demonstrates that our algorithm consistently produces nearly optimal fair ranking, far exceeding the performance suggested by our theoretical bound. Furthermore, the algorithm shows robustness across different metrics, achieving results comparable to or superior to specialized algorithms for the Kendall-tau metric (Chakraborty et al., 2025b).

Improving the approximation factor would be an immediate open question. It would also be interesting to further refine our algorithm's analysis to close the gap between its theoretical and empirical performance. Other promising research directions involve exploring stronger fairness notions than top-$k$ fairness, such as block fairness (Chakraborty et al., 2022). It is also interesting to study fairness in more general settings, such as for overlapping groups or in a group-oblivious setting.

## Acknowledgments

Diptarka Chakraborty was supported in part by an MoE AcRF Tier 1 grant (T1 251RES2303) and a Google South & South-East Asia Research Award. The research of Barna Saha and Arya Mazumdar was supported by NSF TRIPODS award 2217058 (EnCORE Inst.).

## Impact Statement

This paper presents work whose goal is to advance the field of machine learning. There are many potential societal consequences of our work, none of which we feel must be specifically highlighted here.

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

## A. $4$-approximation for Fair Rank Aggregation under the Kendall-tau distance

In this section, we argue that the output ranking of Algorithm 1 attains a 4-approximation for the fair (full) rank aggregation problem under the Kendall-tau distance. For any two rankings $\pi$ and $\sigma$, let the *Kendall-tau distance* between them be denoted by $\kappa(\pi,\sigma)$.

**Theorem A.1** ( (Diaconis & Graham, 1977))**.** *For any two $\pi,\sigma \in \mathcal{S}_d$, $\kappa(\pi,\sigma) \leq F(\pi,\sigma) \leq 2\kappa(\pi,\sigma)$.*

We first prove that an $\alpha$-approximate aggregate ranking to the fair rank aggregation under the Spearman footrule distance is a $2\alpha$-approximation to the fair rank aggregation under the Kendall-tau distance. This, combined with Theorem 4.1, immediately shows that our algorithm finds a 4-approximation to fair rank aggregation under the Kendall-tau distance.

**Theorem A.2.** *Let $S \subseteq S_d$ and $\tilde{\pi}$ be an $\alpha$-approximate fair aggregate ranking for the fair rank aggregation problem under the Spearman footrule distance. Then $\tilde{\pi}$ is a $2\alpha$ approximation for the fair rank aggregation problem under the Kendall-tau distance.*

*Proof.* For the set $S$, let $\pi^*$ be an optimal fair aggregate ranking under the Spearman footrule distance, and $\sigma^*$ be an optimal fair aggregate ranking under the Kendall-tau distance.

As $\tilde{\pi}$ is an $\alpha$-approximate fair aggregate ranking for the fair rank aggregation problem under the Spearman footrule distance,

$$\sum_{\pi \in S} F(\pi,\tilde{\pi}) \leq \alpha \sum_{\pi \in S} F(\pi,\pi^*).$$

By the optimality of $\pi^*$,

$$\sum_{\pi \in S} F(\pi,\pi^*) \leq \sum_{\pi \in S} F(\pi,\sigma^*).$$

Combining these, we obtain

$$\sum_{\pi \in S} F(\pi,\tilde{\pi}) \leq \alpha \sum_{\pi \in S} F(\pi,\sigma^*) \qquad (12)$$

By Theorem A.1, we have that for any ranking $\pi$, $\kappa(\pi,\tilde{\pi}) \leq F(\pi,\tilde{\pi})$. Also by Theorem A.1, we get that $F(\pi,\sigma^*) \leq 2\kappa(\pi,\sigma^*)$. Putting these together, we obtain that

$$\begin{aligned}
\sum_{\pi \in S} \kappa(\pi,\tilde{\pi}) &\leq \sum_{\pi \in S} F(\pi,\tilde{\pi}) \\
&\leq \alpha \sum_{\pi \in S} F(\pi,\sigma^*) \qquad \text{(By Equation 12)} \\
&\leq 2\alpha \sum_{\pi \in S} \kappa(\pi,\sigma^*)
\end{aligned}$$

which concludes the proof. $\qquad\square$

By Theorem 4.1, Algorithm 1 outputs a fair ranking which is a 2-approximation for the fair rank aggregation under the Spearman footrule distance. Therefore, as a direct corollary of Theorem A.2, we get that this fair ranking is a 4-approximation for the fair rank aggregation problem under the Kendall-tau distance.

## B. Alternative $2$-approximation algorithm for Fair Rank Aggregation

In this section, we provide an alternative 2-approximation algorithm for fair rank aggregation. To do that, we again use a two-step process. Before that, we describe the algorithms we will use.

We can also solve the problem of finding an optimal $(\bar{\alpha},\bar{\beta})$-constrained bottom-$k$ ranking. Formally, let a bottom-$k$ ranking $\tau$ be a bijection from a domain $D_\tau \subseteq [d]$ to $\{d-k+1,\cdots,d\}$. Consider a partitioning of $d$ candidates into $g$ groups, $G_1,\cdots,G_g$ and for each group an $\alpha_i,\beta_i \in [0,d]$. For

$\bar{\alpha}=(\alpha_1,\cdots,\alpha_g),\bar{\beta}=(\beta_1,\cdots,\beta_g)$, an $(\bar{\alpha},\bar{\beta})$-constrained bottom-$k$ ranking is one such that, for all $G_i\in\mathcal{G}$, there are at least $\alpha_i$ and at most $\beta_i$ elements from $G_i$ in $D_\tau$. Let the Spearman footrule distance between a ranking and a bottom-$k$ ranking be defined similarly to that between a ranking and a top-$k$ ranking, except that elements not in $D_\tau$ are placed at rank $k$.

We first describe a polynomial time algorithm $\mathcal{B}$ to obtain a $\bar{\alpha},\bar{\beta}$-constrained bottom-$k$ ranking $\tau$, that minimizes that following objective

$$2\sum_{\pi\in S}\sum_{i\in[d]}(\tau(i)-\pi(i))\cdot\mathbb{1}_{\pi(i)<\tau(i)}.$$

Observe that this can be formulated as a linear program which is similar to Fair-K-LP. The objective is now to minimize $\sum_{i\in[d]}\sum_{j>k}w_{ij}x_{ij}$, where $w_{ij}=2\sum_{\pi\in S}(j-\pi_i)\cdot\mathbb{1}_{\pi(i)<j}$. The constraints are similar in form, except that as the variables are now $x_{ij}$ for all $i\in[d],j>k$, the constraints now lie on these variables. It can be verified that the arguments in the proof of Lemma 3.4 hold identically for this linear program. Then, using a proof identical to that of Theorem 3.1, there exists an algorithm that outputs a constrained bottom-$k$ ranking $\tau$ that minimizes the objective in polynomial time. Let $\mathcal{B}$ be this algorithm.

Further, one can also define a problem similar to Definition 4.2, except that the goal is to extend a bottom-$k$ ranking to a full ranking $\sigma$.

**Definition B.1** (Minimum Cost Bottom-$k$ Ranking Completion Problem). Let $S\subseteq S_d$ be given, as well as a bottom-$k$ ranking $\tau$. Define $\tilde{S}_d:=\{\sigma\in S_d\mid\forall a\in D_\tau,\sigma(a)=\tau(a)\}$. Then the minimum cost bottom-$k$ ranking completion problem is to find $\sigma\in\tilde{S}_d$ which minimizes $2\sum_{\pi\in S}\sum_{i\in[d]}(\sigma(i)-\pi(i))\cdot\mathbb{1}_{\pi(i)<\sigma(i)}$.

**Lemma B.2.** *There is an algorithm that finds a ranking $\sigma$ that is an optimal solution to the minimum cost bottom-$k$ ranking completion problem. The algorithm runs in time $O(nd^2+d^3)$.*

*Proof.* A reduction to an instance of minimum cost perfect matching on a complete bipartite graph similar to that in Lemma 4.3 can be applied. $\square$

**Description of the algorithm** The algorithm is similar to that of Section 4. First, we compute for each group $G_i$ the values $\alpha_i':=|G_i|-\lceil\beta_i\cdot k\rceil$ and $\beta_i':=|G_i|-\lfloor\alpha_i\cdot k\rfloor$. Now, apply algorithm $\mathcal{B}$, with input the set of rankings $S$, with parameters $d-k$ and $\alpha'=(\alpha_1',\cdots,\alpha_g'),\beta'=(\beta_1',\cdots,\beta_g')$ to obtain a $(\bar{\alpha}',\bar{\beta}')$-constrained bottom-$d-k$ ranking $\tau$. Then, we extend $\tau$ to a full ranking $\sigma$ using Lemma B.2.

Recall from Proposition 4.4 we have that $F(\pi,\sigma)=2\sum_{i\in[d]}(\sigma(i)-\pi(i))\cdot\mathbb{1}_{\pi(i)<\sigma(i)}$. Therefore, it holds that

$\mathtt{Obj}(\sigma)=2\sum_{\pi\in S}\sum_{i\in[d]}(\sigma(i)-\pi(i))\cdot\mathbb{1}_{\pi(i)<\sigma(i)}$. Let us then define for any ranking $\sigma$

$$\overrightarrow{\mathtt{Obj}}(\sigma_D)=2\sum_{\pi\in S}\sum_{i\in D}(\sigma(i)-\pi(i))\cdot\mathbb{1}_{\pi(i)<\sigma(i)}.$$

Let $\sigma^*$ be any optimal ranking, and as before let $L^*\subseteq[d]$ be the set of elements placed in the top-$k$ ranks of $\sigma^*$ and let $R^*=[d]\setminus L^*$.

**Lemma B.3.** *Consider the ranking $\sigma$ in Algorithm 2. It holds that $\sigma$ is a fair ranking, and $\mathtt{Obj}(\sigma)\leq\overrightarrow{\mathtt{Obj}}(\sigma_{R^*}^*)+\mathtt{OPT}$.*

*Proof.* Recall that the fairness constraints require that for a group $G_i$, there are at least $\lfloor\alpha_i\cdot k\rfloor$ and at most $\lceil\beta_i\cdot k\rceil$ candidates. Observe that an equivalent formulation of the fairness constraint is to require that the bottom $d-k$ ranks contain at least $|G_i|-\lceil\beta_i\cdot k\rceil$ and at most $|G_i|-\lfloor\alpha_i\cdot k\rfloor$ candidates. By the definition of $(\bar{\alpha}',\bar{\beta}')$, $\sigma$ must be a fair ranking.

Then by similar argument as that for Lemma 4.5 we can argue that $\overrightarrow{\mathtt{Obj}}(\sigma_{R'}')\leq\overrightarrow{\mathtt{Obj}}(\sigma_{R^*}^*)$ and that $\mathtt{Obj}(\sigma_{L'}')\leq\overrightarrow{\mathtt{Obj}}(\sigma_{L^*}^*)+\overrightarrow{\mathtt{Obj}}(\sigma_{R^*}^*)$.

Therefore, it holds that

$$\begin{aligned}\mathtt{Obj}(\sigma)&=\overrightarrow{\mathtt{Obj}}(\sigma_{L'})+\overrightarrow{\mathtt{Obj}}(\sigma_{R'})\\&\leq\overrightarrow{\mathtt{Obj}}(\sigma_{L^*}^*)+\overrightarrow{\mathtt{Obj}}(\sigma_{R^*}^*)+\overrightarrow{\mathtt{Obj}}(\sigma_{R^*}^*)\\&\leq\overrightarrow{\mathtt{Obj}}(\sigma_{R^*}^*)+\mathtt{OPT}\end{aligned}$$

$\square$

*Alternative proof of Theorem 4.1.* From Lemma B.3, the ranking $\sigma$ output by Algorithm 2 satisfies $\mathtt{Obj}(\sigma)\leq\overrightarrow{\mathtt{Obj}}(\sigma_{R^*}^*)+\mathtt{OPT}$.

Therefore,

$$\begin{aligned}\mathtt{Obj}(\sigma)&\leq\overrightarrow{\mathtt{Obj}}(\sigma_{R^*}^*)+\mathtt{OPT}\\&\leq\overrightarrow{\mathtt{Obj}}(\sigma_{[d]}^*)+\mathtt{OPT}\\&\leq\mathtt{OPT}+\mathtt{OPT}=2\mathtt{OPT}.\end{aligned}$$

$\square$

# C. Additional details on experiments

**Improved Algorithm** Here, we formally provide Algorithm 3 as implemented in the experiments. Essentially, it runs both Algorithm 1 and Algorithm 2 and returns the better of the two rankings found. Note that the output of this modified algorithm (Algorithm 3) is still a 2-approximation in theory. However, we observe that this algorithm performs better in practice on the Movielens dataset.

**Algorithm 2** FAIR RANK AGGREGATION

1: **Input:** Set of rankings $S$
2: For each group $G_i$, compute $\alpha'_i := |G_i| - \lceil \beta_i \cdot k \rceil$ and $\beta'_i := |G_i| - \lfloor \alpha_i \cdot k \rfloor$.
3: $\tau \leftarrow (\bar{\alpha'}, \bar{\beta'})$-constrained bottom $d-k$ ranking of $S$ using Algorithm $\mathcal{B}$.
4: $\sigma \leftarrow$ full ranking from extending $\tau$ with $S$ using the algorithm of Lemma B.2.
5: Return $\sigma$

---

**Algorithm 3** FAIR RANK AGGREGATION

1: **Input:** Set of rankings $S$
2: $\tau \leftarrow$ top-$k$ fair ranking of $S$ from applying algorithm $\mathcal{A}$.
3: $\sigma \leftarrow$ full ranking from using Lemma 4.3 with $S$ and $\tau$ as input.
4: For each group $G_i$, compute $\alpha'_i := |G_i| - \lceil \beta_i \cdot k \rceil$ and $\beta'_i := |G_i| - \lfloor \alpha_i \cdot k \rfloor$.
5: $\tau' \leftarrow (\bar{\alpha'}, \bar{\beta'})$-constrained bottom $d-k$ ranking of $S$ using Algorithm $\mathcal{B}$.
6: $\sigma' \leftarrow$ full ranking from extending $\tau'$ with $S$ using the algorithm of Lemma B.2.
7: **if** $\mathtt{Obj}(\sigma) \leq \mathtt{Obj}(\sigma')$ **then**
8:     Return $\sigma$
9: **else**
10:     Return $\sigma'$
11: **end if**

---

## D. Hardness results

It is known that the problem of rank aggregation under Spearman footrule distance (without fairness) can be solved exactly in polynomial time by a reduction to an instance of minimum cost perfect matching (Dwork et al., 2001a). In this section, we show that a similar approach is unlikely to provide an exact algorithm for the problem of fair rank aggregation under Spearman footrule distance.

First, let us define the following problem.

**Definition D.1** (Fair Colorful Weighted Perfect Matching). Given $G = (V \cup W, E)$ be a weighted complete bipartite graph with bipartition $V, W$, a subset $Z \subset W$, and a function *col* mapping vertices of $V$ to a set of colors $C = \{c_1, \cdots, c_g\}$. Further, for each color $c_i$ let $\alpha_i, \beta_i \in [0, 1]$ be given, with $\bar{\alpha} = (\alpha_1, \cdots, \alpha_g), \bar{\beta} = (\beta_1, \cdots, \beta_g)$, and a target value $t$. Then the objective is to find a perfect matching $M$ of value at least $t$ such that the following constraint holds: $\forall c_i \in C$, $\alpha_i \cdot |Z| \leq |\{e = (u, v) \in M \mid col(u) = C_i, v \in W\}| \leq \beta_i \cdot |Z|$.

An instance of fair rank aggregation can be reduced to an instance of the maximization version of fair colorful weighted perfect matching. Construct the vertex set $V$ by creating a vertex $v_i$ for each $i \in [d]$. Construct the vertex set $W$ by creating a vertex $w_j$ for each $j \in [d]$. Let $Z$ the set

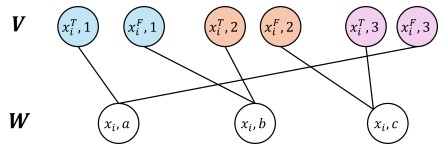

*Figure 6.* Illustration of the variable gadget in the reduction for a variable $x_i$. Drawn edges are of weight 1, and edges of weight 0 are omitted. Note that the three colors in the reduction are distinct from the colors used for other variables.

$\{w_1, \cdots, w_k\}$. The weight of an edge $e = (v_i, w_j)$ for $v_i \in V$ and $w_j \in W$ is set to be $-\sum_{\pi \in S} |\pi(i) - j|$. The fairness parameters are exactly the $\bar{\alpha}, \bar{\beta}$ as in fair rank aggregation. Note that the edge weights are negative, as rank aggregation seeks to minimize cost, and fair colorful weighted perfect matching aims to maximize cost. Given a solution (i.e., a matching) to the fair colorful weighted perfect matching problem, if a vertex $v_i$ is matched to $w_j$, then place element $i$ at rank $j$ in $\sigma$. One can observe that if the matching has cost $-k$, then the ranking $\sigma$ has $\mathtt{Obj}(\sigma) = k$, thus the reduction clearly holds.

We prove that the problem of fair colorful weighted perfect matching is, in fact, NP-hard. Therefore, we cannot hope to solve fair rank aggregation by solving this matching problem, or, for instance, formulating it as a linear program similar to Fair-K-LP.

**Theorem D.2.** *Fair Colorful Weighted Perfect Matching is* NP*-Hard.*

Our reduction will be via a variant of 3-SAT, where each variable appears in at most three clauses, which is known to be NP-Hard (Tovey, 1984). We formally define it below.

**Definition D.3** ((3,3)-SAT). Consider a boolean formula $B = C_1 \vee C_2 \vee \cdots \vee C_m$ on variables $x_1, \cdots, x_n$ where each clause contains at most three variables, and each variable appears in at most three clauses. Is the formula $B$ satisfiable?

**Theorem D.4** ((Tovey, 1984)). *(3,3)-SAT is* NP*-Hard.*

We now describe the reduction, constructing the vertex sets $V, W$ and the weighted edge set $E$.

We construct a 'variable gadget' as follows. For each variable $x_i$, add 6 vertices to $V$ and 3 vertices to $W$. Let the vertices added to $V$ be labeled $(x_i^T, 1), (x_i^F, 1), (x_i^T, 2), (x_i^F, 2), (x_i^T, 3), (x_i^F, 3)$. Let the vertices added to $W$ be labeled $(x_i, a), (x_i, b), (x_i, c)$. Further, add three unique colors $c_{i,1}, c_{i,2}, c_{i,3}$, where $(x_i^T, j), (x_i^F, j)$ are mapped to color $c_{i,j}$ for each $j \in \{1, 2, 3\}$. Finally, add 6 edges of weight 1 as follows. $(x_i, a)$ has an edge to $(x_i^T, 1)$ and $(x_i^F, 3)$. $(x_i, b)$ has an edge to $(x_i^T, 2)$ and $(x_i^F, 1)$. $(x_i, c)$ has an edge to $(x_i^T, 3)$ and $(x_i^F, 2)$. Further, let the three vertices added to $W$ also be in the subset $Z$. See Figure 6 for an illustration of how the edges are constructed.

Next, we construct a 'clause gadget' as follows. Take any arbitrary ordering of the clauses. For each clause $C_i$, add a

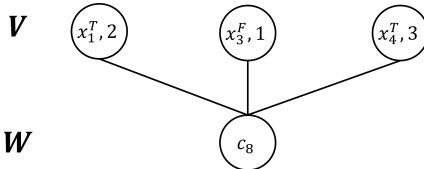

*Figure 7.* Illustration of the clause gadget in the reduction for a clause $C_8 = x_1 \vee \bar{x}_3 \vee x_4$. Drawn edges are of weight 1, and edges of weight 0 are omitted. Colors omitted as they are not relevant for the clause gadget.

vertex $C_i$ to $W$. For each variable $x_j$ in $C_i$, suppose this is the $r$-th appearance of the variable in the formula. If $x_j$ appears non-negated, then construct an edge of weight 1 from vertex $(x_j^T, r)$ to vertex $C_i$. If $x_j$ appears negated, then instead construct an edge of weight 1 from vertex $(x_j^T, r)$ to vertex $C_i$.

Finally, we add dummy vertices to $W$ until it is of the same cardinality as $V$. To ensure the graph constructed is a complete bipartite graph, let all other edges that are not given a weight in the construction so far be given a weight of 0. Next, let the target value $t$ in the instance of matching be set to $3n + m$. Lastly, we set the color constraints as $\alpha_i = \beta_i = 1/3n$ for all $i \in [g]$. That is, exactly one vertex of each color in $V$ can be matched to vertices of $Z$.

Now, we argue about the correctness of the reduction. We first show the YES instance of $(3,3)$-SAT correctly maps to a YES instance of fair colorful weighted perfect matching.

**Lemma D.5.** *If the instance of $(3,3)$-SAT is satisfiable, then the constructed instance of maximum cost perfect matching is satisfiable.*

*Proof.* Construct a matching $M$ as follows. If a variable $x_i$ is set to True in the satisfying assignment of $(3,3)$-SAT, then match $(x_i^F, 1)$ with $(x_i, b)$, match $(x_i^F, 2)$ with $(x_i, c)$ and match $(x_i^F, 3)$ with $(x_i, a)$. If $x_i$ is set to False, then match $(x_i^T, 1)$ with $(x_i, a)$, match $(x_i^T, 2)$ with $(x_i, b)$ and match $(x_i^T, 3)$ with $(x_i, c)$. Observe that the matching constructed indeed satisfies the fairness constraints.

Consider any clause $c_j$, which must be satisfied by at least one variable. Pick any one of the variables that satisfies it, and let it be $x_i$. By the construction, the vertex $c_j$ must have an edge of weight one to some vertex corresponding to $x_i$,

and further, this vertex is not yet part of a matching in $M$. Thus, these two vertices can be matched, adding a cost of 1.

Finally, complete the perfect matching by arbitrarily matching the remaining vertices of $V$ and $W$.

It is clear that the matching satisfies the color constraints on $Z$ and achieves a cost of $3m + n$, showing the constructed instance is satisfiable. $\square$

We now prove that a NO instance of 3-SAT correctly maps to a NO instance of fair maximum cost perfect matching.

**Lemma D.6.** *If the constructed instance of fair maximum cost perfect matching is satisfiable, then the 3-SAT instance is satisfiable.*

*Proof.* Consider a satisfying matching $M$ to the constructed instance of fair maximum cost perfect matching. Observe that only $3m + n$ vertices of $W$ have an edge of weight 1 incident on them. Therefore, in the matching $M$, all of those vertices must be matched using an edge of weight 1.

Fix some $x_i$ and consider its corresponding vertices of $W$, namely $(x_i, a), (x_i, b), (x_i, c)$. One can exhaustively check that the only possible ways of matching which ensure all three vertices are matched using an edge of cost 1, while satisfying the fairness constraints is matching them to all of $(x_i^T, 1), (x_i^T, 2), (x_i^T, 3)$ or to all of $(x_i^F, 1), (x_i^F, 2), (x_i^F, 3)$. If the matching is to $(x_i^T, 1), (x_i^T, 2), (x_i^T, 3)$, then in the $(3,3)$-SAT instance set $x_i$ to False. Else, in the $(3,3)$-SAT instance, set $x_i$ to True. This gives an assignment of the variables.

Now, take any clause $c_j$. Suppose that the vertex $c_j$ is matched to vertex $(x_i^T, j)$ for $j \in \{1, 2, 3\}$. Then by the above argument, it must hold that the vertices $(x_i^F, 1), (x_i^F, 2), (x_i^F, 3)$ must be matched to $(x_i, a), (x_i, b), (x_i, c)$, and so the assignment has $x_i$ set to True, thus satisfying $c_j$. A similar argument holds if $c_j$ is matched to a vertex $(x_i^F, j)$ for $j \in \{1,2,3\}$ instead.

Therefore, the $(3, 3)$-SAT instance is satisfied by the assignment. $\square$

*Proof of Theorem D.2.* It is straightforward to see that the reduction runs in polynomial time, and combined with Lemma D.5 and Lemma D.6, it completes the proof. $\square$

