# OpenReview forum: "Fairness in Aggregation: Optimal Top-$k$ and Improved Full Ranking"
_ICML.cc/2026/Conference — ICML 2026 regular_

### Official Review · Reviewer_mqih · 2026-03-09

**Soundness:** 3
**Presentation:** 3
**Significance:** 3
**Originality:** 3
**Overall Recommendation:** 4
**Confidence:** 2

**Summary:**

This paper investigates the problem of rank aggregation under fairness constraints, primarily utilizing the Spearman footrule distance as the metric. The work explores the challenge of maintaining proportional diversity in the final output, a relevant problem studied by this study in the context of high-stakes ranking systems.

The main contributions include:
    An optimal polynomial-time algorithm for the fair top-k ranking problem, achieved by proving the Total Unimodularity of the constraint matrix in an LP relaxation.
    A 2-approximation algorithm for the fair rank aggregation problem, improving upon the existing 3-approximation state-of-the-art.
    A proof of the NP-hardness of the fair matching problem under the Spearman footrule.
    Experimental results on real-world datasets that corroborate the theoretical findings.

**Compliance With Llm Reviewing Policy:**

Affirmed.

**Final Justification:**

I will keep my score.

**Key Questions For Authors:**

In the proof of Lemma 3.3, would the TU property still hold if more complex fairness constraints were introduced, such as overlapping groups or multi-level hierarchical constraints?

The algorithm relies on the Ellipsoid method or Gurobi as an ILP solver. How does the practical wall-clock time scale when the number of candidates reaches tens of thousands?

Is there theoretical evidence suggesting that a 2-approximation is the lower bound for the fair full rank aggregation problem, or is there potential for an exact polynomial-time algorithm in the future?

**Limitations:**

Yes

**Strengths And Weaknesses:**

# Strengths

The paper makes substantial progress in the theoretical understanding of fair ranking. Specifically, the use of Total Unimodularity to prove the optimality of the fair top-k problem is an elegant and efficient approach. Improving the approximation bound from 3 to 2 for the full rank aggregation problem is a non-trivial advancement in this field.

In real-world scenarios such as hiring, recommendation systems, and search engines, the top-k results are often the only ones that matter to stakeholders, making an optimal algorithm for this specific variant highly impactful.

The work provides a comprehensive treatment of the topic, spanning from algorithm design and rigorous proofs to complexity analysis and empirical validation, forming a solid research loop.

# Weaknesses

The experiments utilize the Movielens dataset and a Football dataset. While these are standard benchmarks, the paper lacks scalability tests on instances with a much larger number of candidates or a larger volume of input rankings, which would be more representative of modern ML applications.

The paper employs hard proportional constraints. In cases where the input rankings are extremely skewed, these hard constraints might lead to a significant increase in the objective value. A discussion on the impact of relaxing these constraints would have been beneficial.

While the Spearman footrule is computationally advantageous, the Kendall-tau distance is often considered more natural for rank aggregation. Although the authors discuss robustness and provide a 4-approximation corollary for Kendall-tau, a deeper discussion on the generalizability of the core algorithms to other metrics is somewhat lacking.

---

> ### Author Rebuttal · Authors · 2026-03-31
>
> Thanks for the review and positive feedback. Let us provide a brief clarification on the questions raised.
>
> 1. The TU property does not naturally extend to an arbitrary notion of fairness. Proving the constraint matrix is TU will require studying the matrix formed by those constraints, and the analysis and proof may differ. In fact, in the case of overlapping groups, this TU property does not hold in general. While it is a natural extension to consider when candidates belong to multiple groups, it is worth noting that [Celis, Straszak, Vishnoi, ICALP 2018] showed that the problem of simply determining if a fair ranking exists is NP-Hard when a candidate can belong to at least 3 groups; therefore, the problem becomes significantly more challenging in this setting. Extending our approach to multi-level hierarchical constraints depends on whether it can be formulated as an LP, and then on the constraint matrix.
>
> 2. In practice, when $d = 10000$, the wall-clock time of Algorithm 5 is about 700 seconds, and the ILP takes about 1500 seconds, using a hardware configuration with an AMD EPYC 7763 CPU and 512GB of RAM. Note the different hardware configuration from the experiments in the paper, as Gurobi has significant memory usage from having a large set of candidates. We note that while in theory we require the ellipsoid method to establish the polynomial time complexity, in practice, solvers can exploit parallelism to run both the ellipsoid and simplex algorithms..
>
> 3. To the best of our knowledge, there is no hardness of approximation result known for the fair rank aggregation under the Spearman footrule distance; in fact, we do not know any such hardness with respect to any well-known distance metric. We note that we show in the appendix that a fair variant of the matching problem, even on a complete bipartite graph -- a generalization of the fair rank aggregation under Spearman footrule -- is already NP-hard. However, that does not directly refute the possibility of having a better approximation/exact algorithm for the fair rank aggregation under the Spearman footrule. We identify this as an interesting theoretical question to explore.

---

> > ### Author Rebuttal · Reviewer_mqih · 2026-04-04
> >
> > My concerns have been adequately addressed.  This paper appears to be well-written. Unfortunately, I am not an expert in this field and cannot provide further professional comments.

---

### Official Review · Reviewer_QFHZ · 2026-03-09

**Soundness:** 2
**Presentation:** 2
**Significance:** 3
**Originality:** 3
**Overall Recommendation:** 5
**Confidence:** 4

**Summary:**

The authors address the problem of rank aggregation by incorporating constraints into the optimization problem to ensure fair representations. Specifically, they study two fairness-aware variants under the Spearman footrule distance.
First, since the requirements of some applications focus only on the top-k ranked items (such as hiring processes, where the number of available positions is limited) the authors consider the problem of computing a fair aggregate top-k ranking and present the first known optimal algorithm for this problem. To this end, they first formulate the problem as an Integer Linear Program (ILP) and analyze its Linear Programming (LP) relaxation. By studying the structure of the constraint matrix, they show that it is totally unimodular, which implies that the LP admits an optimal integral solution and can therefore be solved directly without requiring additional integer programming techniques.
Second, they study the fair full-rank aggregation problem across all candidates. In this setting, they make significant progress by providing a 2-approximation algorithm, improving upon the previously known 3-approximation result in the literature.
The theoretical contributions are complemented with empirical evaluations on two real-world datasets.

**Compliance With Llm Reviewing Policy:**

Affirmed.

**Final Justification:**

The authors adequately addressed my main concerns in the rebuttal, clarifying that several of the issues I raised regarding the fairness interpretation of the full-ranking setting are inherent to the standard problem formulation adopted in prior work. While I still believe these limitations should be kept in mind when interpreting the practical implications of the results, they do not detract from the technical validity of the contributions.

Overall, the paper presents meaningful theoretical advances on the fair rank aggregation problem, particularly through the exact polynomial-time solution for the top-k setting and the improved approximation guarantee for the full-ranking variant. I therefore view the paper as a solid contribution within the scope of the formulation considered.

**Key Questions For Authors:**

1. Did the authors explore an iterative block-based completion approach to maintain fairness guarantees at multiple cut-off points in the ranking? What would be the main obstacle to this alternative, either from a theoretical or computational perspective?

2. Could the authors clarify how the resulting complete ranking can still be considered a “globally fair” (fair full ranking) if the candidates outside the top-k are ordered only by minimizing the cost/distance, without any representation constraints?

3. Does the 2-approximation guarantee hold only because the list completion step is done without fairness constraints (i.e., unconstrained)? If quotas were applied to the whole ranking (e.g., fair completion), what theoretical bound could be guaranteed?

**Limitations:**

yes

**Strengths And Weaknesses:**

Strengths
In my opinion, the paper has several strong points. The main technical contribution is the identification of a totally unimodular (TU) matrix in the formulation of the problem. By showing that the top-k problem with representation quotas can be written with a TU matrix, the authors make it possible to solve the problem through its linear programming (LP) relaxation in polynomial time. Because of this property, the LP solution is guaranteed to be integral, which means that the method can obtain the exact optimal solution for the fair top-k aggregation problem. This result helps close an important computational gap in the literature.
Another important strength is the use of a “truncated” Spearman distance. In this metric, the real positions are used for the candidates in the top-k, while all candidates outside the top-k are “assigned” to position k+1. In this way, the penalty mainly depends on the top part of the ranking. This is a reasonable design choice and works well for real applications such as search engines or recommender systems, where the order of the tail of the ranking usually has little importance for users. Finally, the paper also improves the theoretical bound for the full ranking problem. In particular, it reduces the approximation ratio from a 3-approximation, which was the previous state of the art, to a 2-approximation. This can be considered a solid theoretical contribution, since achieving this type of improvement is usually quite difficult in geometric aggregation problems.

Weaknesses
In my opinion, the paper also has some important limitations that should be discussed more clearly. First, the paper gives the impression that the proposed method solves the fair aggregation problem for the full ranking (the authors state in lines 30–31: “Second, we consider fair (full) rank aggregation over all candidates (not specifically on top-k).”). However, when reading carefully how the authors construct the complete ranking, it becomes clear that this is not really the case. Their method produces a perfectly fair top-k ranking, but the rest of the list is essentially obtained through a completion step. In practice, the paper presents a “fair top-k with an optimized tail” as if it were a fully fair ranking for all candidates. This issue becomes clearer when looking at the rule used to fill the rest of the ranking. After selecting the fair top-k, the algorithm simply fixes these candidates and then orders the remaining ones by minimizing the cost with respect to the original input rankings. In this second step, no diversity or representation constraints are imposed. As a result, the fairness requirements that are carefully enforced in the top-k are completely abandoned for the rest of the list.
This design choice is also closely related to the main theoretical improvement claimed in the paper. The authors highlight that they improve the approximation bound from a 3-approximation, which was the previous state of the art, to a 2-approximation. However, this improvement is achieved because the fairness constraints are no longer enforced after position k. Once these constraints are removed, it becomes much easier for the algorithm to minimize the ranking distance and obtain a better approximation bound. In this sense, the improvement in the theoretical guarantee comes at the cost of sacrificing fairness in most of the ranking.
Another limitation appears when considering realistic usage scenarios, such as paginated search results. In practice, users do not only look at the first page of results. With the proposed approach, the first page (the top-k) would satisfy the fairness constraints, but the following pages would contain candidates ordered without any fairness guarantees. In applications such as search engines or recommendation systems, a block-based or iterative approach that enforces fairness page by page may be more appropriate to maintain fairness across the entire ranking.

---

> ### Author Rebuttal · Authors · 2026-03-31
>
> We thank the reviewer for the feedback and also for appreciating our theoretical contribution on improving the approximation factor. Let us clarify a few points.
>
> First, we indeed provide a **2-approximation for the fair full rank aggregation** problem, improving over the known 3-approximation. We would like to clarify that the definition of fair ranking we use as in Definition 2.1 (where the upper and lower bound constraints are on the top-k candidates) is exactly the same as that in previous work [Chakraborty, Das, Dey, Yan, IJCAI 2025] (refer to Definition 1 there) and [Chakraborty, Das, Khan, Subramanian, NeurIPS 2022] (refer to Definition 2.5 there). In all of these definitions, the upper and lower bound (fairness) constraints are on the number of candidates of each group placed in the top-$k$ positions; there are no constraints on the remaining positions (beyond $k$). Under this notion of fair ranking, the fair (full) rank aggregation problem asks to output a full ranking. The best known result for exactly this variant of the problem, i.e., fair (full) rank aggregation under Spearman footrule distance, is a 3-approximation [Chakraborty, Das, Khan, Subramanian, NeurIPS 2022].
>
>
>
> Next (in reply to points 1 and 3), we acknowledge that in some applications, a stronger notion of fairness may be desired that considers more than just the top-$k$, and the reviewer's suggestion to consider fairness constraints across multiple cut-offs could be useful. Indeed, our algorithm can be extended to an iterative approach to find a ranking that meets these stronger fairness constraints block-wise. However, our current analysis for the 2-approximation guarantee does not immediately generalize to allow bounding the cost of the output ranking of this iterative version of our algorithm compared to an optimal block-fair ranking. We have already posed the problem of studying such other fairness notions for rank aggregation as an interesting open direction.

---

> > ### Author Rebuttal · Reviewer_QFHZ · 2026-04-04
> >
> > We thank the authors for their thoughtful response. All my doubts have been clarified and, as such, will keep the positive score.

---

### Official Review · Reviewer_UV4m · 2026-03-10

**Soundness:** 4
**Presentation:** 3
**Significance:** 4
**Originality:** 3
**Overall Recommendation:** 5
**Confidence:** 3

**Summary:**

The paper studies fair rank aggregation.
Here we are given $n$ permutations (rankings) of the set $\{1, …, d\}$ of objects.
We are also given a partition of the objects into groups $G_1, …, G_t$ and a number $1 ≤ k ≤ d$.
We must aggregate the rankings into a single ranking such that
the top $k$ items are distributed somewhat equally across the groups.

Here fairness is a constraint, and the sum of the distances of the ranking we output
from the input rankings is the objective, which we must minimize.
When we only have to select $k$ objects and rank them,
the paper gives a polynomial-time algorithm.
When we must rank all objects, the paper gives a 2-approximation algorithm,
which improves upon the previous 3-approximation algorithm.

**Compliance With Llm Reviewing Policy:**

Affirmed.

**Final Justification:**

I didn't ask any questions. My evaluation of the paper remains the same.

**Key Questions For Authors:**

none

**Limitations:**

yes

**Strengths And Weaknesses:**

The paper studies an important and fundamental problem,
and improves the state-of-the-art approximation factor.
I consider this to be a significant result.

The paper is mostly well-written, except that I initially found Theorem 1.2 slightly confusing
because there was no mention of $k$ being part of the input.
(I suggest the authors to explicitly mention something like "we compute a full ranking,
but the fairness of the aggregated ranking is determined by the first $k$ entries."
when they introduce this problem.)

The analysis of the theoretical results uses elementary techniques and is easy to understand, which is commendable.

The algorithm also seems to perform well in practice,
doing better than what the theoretical analysis would predict.

---

> ### Author Rebuttal · Authors · 2026-03-31
>
> We thank the reviewer for their positive feedback and for appreciating our work. We will incorporate the suggestion in the final version.

---

> > ### Author Rebuttal · Reviewer_UV4m · 2026-04-01
> >
> > NA

---

### Official Review · Reviewer_JyBW · 2026-03-13

**Soundness:** 1
**Presentation:** 2
**Significance:** 1
**Originality:** 3
**Overall Recommendation:** 2
**Confidence:** 4

**Summary:**

This work gives an algorithm for top-k rank aggregation under minority protection constraints. In particular, they aim to find a ranking which minimizes the Spearman footrule distance between the output ranking and all input rankings subject to two types of constraints: the first constraint specifies a minimum representation level for a protected group in the output ranking, and the second specifies a maximum representation level for a dispreferred group. They introduce a linear programming formulation for outputting an optimal top-k ranking subject to the two fairness constraints.

**Compliance With Llm Reviewing Policy:**

Affirmed.

**Key Questions For Authors:**

1. Why is rank aggregation using the footrule metric appropriate for high stakes settings? I tend to view this as appropriate for e.g. database settings due to its computational efficiency, but it fails to satisfy socially desirable criteria, notably the Condorcet criterion.

**Limitations:**

Yes

**Strengths And Weaknesses:**

I did not find this work to be well motivated. The introduction positions this algorithm as appropriate for high stakes settings such as hiring and college admissions, but the aggregation method they propose clearly fails to be Condorcet; thus it may fail to even shortlist a clear consensus winner.

Leaving aside the motivation, this paper has clear technical issues. First, they reference a well-known result by Diaconis and Graham which states that the Kendall-tau distance and the Spearman distance are within a factor two of each other, and use this to claim that their algorithm solves the fairness constrained rank aggregation problem under the Kendall-tau distance with the same approximation. It is not clear why the approximation factor should be preserved when the fairness constraint is added.

More seriously, the two main technical results of the paper (theorem 3.1 and the hardness reduction presented in appendix F) are inconsistent with each other. Theorem 3.1 states that the top-k fair rank aggregation problem can be solved exactly by linear programming for all values of k, including full rankings, but this contradicts the claim that the fair rank aggregation problem for full rankings in NP-hard. One would expect to see a threshold on the value of k beyond which the aggregation problem becomes hard, e.g. the problem is easy when k is a small constant, but hard when k = O(d).

---

> ### Author Rebuttal · Authors · 2026-03-31
>
> We thank the reviewer for their feedback. We understand that there has been some misunderstanding regarding our contribution. Let us clarify them below.
>
> First of all, we assume the reviewer is referring to the claim in lines 122-124 in the submitted paper, where we say, ``Though as an immediate corollary of our theoretical result, we get a 4-approximation for the fair rank aggregation under the Kendall-tau metric''. Our algorithm **indeed provides a 4-approximation** guarantee for the fair rank aggregation problem under the Kendall tau distance. To see this, first note that Diaconis and Graham showed that for any two rankings (irrespective of whether they are fair or not) $\pi, \sigma$, it holds that $F(\pi, \sigma) \le \kappa(\pi, \sigma) \le 2F(\pi, \sigma)$, where $\kappa$ denotes the Kendall tau distance and $F$ denotes the Spearman footrule distance. Let $\sigma^\*$ be an optimal fair ranking under the Kendall-tau metric, $\pi^*$ be an optimal fair ranking under the Spearman footrule distance, and $\tilde{\pi}$ be the output of our algorithm.
>
> Then, it holds that $\sum_{\pi \in S} F(\pi, \tilde{\pi}) \le 2 \sum_{\pi \in S} F(\pi, \pi^\*)$ (since our algorithm's output is a 2-approximation for Spearman footrule). Since $\pi^\*$ is an optimal fair aggregate ranking under Spearman footrule, it holds that $\sum_{\pi \in S} F(\pi, \pi^\*) \le \sum_{\pi \in S} F(\pi, \sigma^\*)$. Then, by applying the inequalities proven by Diaconis and Graham, we get that $\sum_{\pi \in S} \kappa(\pi, \tilde{\pi}) \le 2 \sum_{\pi \in S} F(\pi, \tilde{\pi}) \le 4 \sum_{\pi \in S} \kappa(\pi, \sigma^*)$. Thus, the objective value of $\tilde{\pi}$ is within a 4-factor of the objective value of an optimal fair aggregate ranking under Kendall tau distance. As the fairness notions are identical, the ranking output by our algorithm is still a valid fair ranking for fair rank aggregation under the Kendall tau distance. To avoid any confusion, we plan to include the above argument in the appendix of the final version. We emphasize that, although theoretically our algorithm gives only a 4-approximation to the fair rank aggregation under the Kendall tau distance (as a corollary, as explained above), our empirical findings establish that it is comparable (and in some cases even outperform) to the best known fair rank aggregation algorithm with respect to the Kendall-tau metric due to [Chakraborty, Das, Dey, Yan, IJCAI 2025].
>
> Second, Theorem 3.1 and the hardness result in Appendix F are **not contradictory**. We emphasize that our result in Theorem 3.1 applies to the problem where the output is a top-$k$ ranking, not a full ranking. Thus, we essentially solve the problem stated in Definition 2.3 exactly in polynomial time. In contrast, our hardness result is for a generalized version of the problem of finding an output full ranking with fairness. In fact, Theorem 3.1 and the hardness in Appendix F provide a clear separation between the computational complexity for these two problems, providing evidence that while finding an optimal top-$k$ aggregate list (with fairness) is poly-time solvable, it might be hard to find an optimal full ranking (with the same fairness criteria).
>
>
>
> In terms of **motivation**, beyond its use in databases and information retrieval such as in [Fagin, Kumar, Mahdian, Sivakumar, Vee, PODS 2004]and [Farah, Vanderpooten, SIGIR 2007], the fair rank aggregation under Spearman footrule distance finds applications in other domains -- such as recommender systems [Baltrunas, Makcinskas, Ricci, RecSys 2010], [Oliviera, Diniz, Lacerda, Merschmanm, Pappa, ACM TIST 2020] to avoid bias in the top recommendations shown to users, as well as in aggregating search [Bar-Ilan, Levene, Mat-Hassan, Computer networks 2006], [Sculley, SDM 2007]. The fair rank aggregation under Spearman footrule distance has also been studied previously by [Chakraborty, Das, Khan, Subramanian, NeurIPS 2022] and [Wei, Islam, Schieber, Basu Roy, SIGMOD 2022]. We agree that, in settings where the winner (only the top-most ranked candidate) is important, the Spearman footrule distance may not be as preferable due to not fulfilling the Condorcet criteria.

---

> > ### Author Rebuttal · Reviewer_JyBW · 2026-04-04
> >
> > Thank you for the clarification. I am now on board with the claimed 4-approximation. However, I maintain that theorem 3.1 as currently stated is inconsistent with the hardness result in appendix F. The algorithm is claimed to be efficient for all values of k, including for a full ranking of the d candidates. If this result requires that k is a small constant, this should be reflected in the theorem statement.

---

> > > ### Author Response · Authors · 2026-04-04
> > >
> > > Thanks for asking this question. Let us clarify. Note that in the Fair Top-$k$ Rank Aggregation problem (Definition 2.3), the parameter $k$ for the desired top-$k$ list (that is, to output) and the $k$-length prefix on which upper and lower bound (fairness) constraints are imposed, are exactly the same. That makes the problem computationally efficient, and this is captured in Theorem 3.1. Once we want a top-$k'$ list to output ($k' = d$ for the full rank aggregation), but the upper and lower bound (fairness) constraints are imposed on the top-$k$ prefix (where $k $ need not be equal to $k'$), then the problem starts becoming hard, and we provide that evidence in the hardness result in Appendix F. In this explanation, we introduce the new parameter $k'$ in the hope of providing better clarification. We emphasize that, as such, for Theorem 3.1, we \textbf{do not} need $k$ to be a small constant, and indeed, for any arbitrary value of $k$, the Fair Top-$k$ Rank Aggregation problem (Definition 2.3) is efficiently solvable (Theorem 3.1).
> > >
> > > To clarify more, if we consider the special case of $k=k'=d$, then the problem essentially boils down to the problem of rank aggregation (without any fairness constraint) under Spearman-footrule, and thus can be solved in polynomial time [Dwork, Kumar, Naor, Sivakumar, WWW 2002]. To see this, for $k=d$, one first needs to check whether the fairness parameters $\bar{\alpha},\bar{\beta}$ (as in Definition 2.1) leads to a feasible fair ranking, and if so, then note any ranking is fair because the upper and lower bound fairness constraints are imposed on the whole $d$-length ranking. This step can trivially be performed in linear time. Now, once we know any ranking is fair, then it is essentially just the rank aggregation without explicitly considering fairness constraints. Thus, taking $k=d$ only trivializes the problem.
> > >
> > > For the fair full rank aggregation (by definition, then $k'=d$, and $k$ is arbitrary, not necessarily be equal to $k'$), we show a 2-approximation in Theorem 4.1, which is an improvement over the 3-approximation from prior works [Chakraborty, Das, Khan, Subramanian, NeurIPS 2022], and as highlighted by another reviewer that such improvements are usually difficult to achieve for this type of geometric median problem (note, the fair full rank aggregation is the geometric median problem over Spearman-footrule metric with even additional fairness constraints). Again, for this result, we \textbf{do not} need $k$ to be a small constant.
> > >
> > > Hope it clarifies your doubt, and if you have any further question please let us know. We would be very happy to include the above explanation in the final version if the reviewer thinks that it provides a better understanding.

---

### Decision · Program_Chairs · 2026-04-30

**Decision:**

Accept (regular)

**Comment:**

The paper looks at the problem of fairness in rankings aggregation under two settings: top-k and full rank aggregation. For both settings, the paper makes theoretical progress:
- it provides an optimal algorithm for top-k
- it improves from a 3- to a 2- approximation for full rank (where the problem is known to be NP-hard).

The proposed algorithms include novel ideas and improve on the state of the art and all reviewers found this to be a significant theoretical contribution and I agree with that assessment. Some of the ideas and proofs are quite simple but this is to be taken more as a positive point. The paper is also well written and the problem is well motivated. The paper is also complemented by experiments.

One reviewer gave a negative score because they has a doubt about the relationship between Thm 3.1 and the hardness results of App F, suspecting a possible contradiction. However, the authors convincingly explained in the rebuttal that there is no contradiction as these result consider different problems. I strongly encourage the authors to carefully adjust the phrasing in their revision to avoid this confusion.

Some extensions to stronger fairness notions would strengthen the results, but those are also not done in related work.

Overall, the paper presents a solid theoretical contribution on an important problem so I recommend to accept it.